# Coupling Experts and Routers in Mixture-of-Experts via an Auxiliary Loss

**Ang Lv**[1,2*], **Jin Ma**[2], **Yiyuan Ma**[2†], **Siyuan Qiao**[2]
[1]Renmin University of China, GSAI  [2]Bytedance Seed

## Abstract

Mixture-of-Experts (MoE) models lack explicit constraints to ensure the router's decisions align well with the experts' capabilities, which ultimately limits model performance. To address this, we propose expert-router coupling (ERC) loss, a lightweight auxiliary loss that tightly couples the router's decisions with expert capabilities. Our approach treats each expert's router embedding as a proxy token for the tokens assigned to that expert, and feeds perturbed router embeddings through the experts to obtain intermediate activations. The ERC loss enforces two constraints on these activations: (1) Each expert must exhibit higher activation for its own proxy token than for the proxy tokens of any other expert. (2) Each proxy token must elicit stronger activation from its corresponding expert than from any other expert. These constraints jointly ensure that each router embedding faithfully represents its corresponding expert's capability, while each expert specializes in processing the tokens actually routed to it. The ERC loss is computationally efficient, operating only on $n^2$ activations, where $n$ is the number of experts. This represents a fixed cost independent of batch size, unlike prior coupling methods that scale with the number of tokens (often millions per batch). Through pre-training MoE-LLMs ranging from 3B to 15B parameters and extensive analysis on trillions of tokens, we demonstrate the effectiveness of the ERC loss. Moreover, the ERC loss offers flexible control and quantitative tracking of expert specialization levels during training, providing valuable insights into MoEs.

## 1 Introduction

Mixture-of-Experts (MoE, Shazeer et al., 2017; Fedus et al., 2022; Lepikhin et al., 2021; Zoph et al., 2022) is a core architecture in modern large language models (LLMs). In MoE models, the feed-forward layer is split into multiple small, specialized "experts." A linear classifier, known as the "router," selects which experts process each input token. By activating a few experts per token, MoE balances efficiency with scaled parameter counts, enabling the training of trillion-parameter models.

Ideally, a router should possess an accurate representation of each expert's capabilities to enable effective token routing. However, traditional MoEs offer no explicit constraints to guarantee this. Without direct access to expert parameters (and therefore their true capabilities), routers resort to trial-and-error learning of routing strategies, often resulting in misrouted tokens whose gradients interfere with expert specialization. While some methods (Lv et al., 2025; Pham et al., 2024) incorporated all experts' activations for routing guidance, they incur substantial computational and memory costs due to denser activation. A lightweight and effective solution to better couple routing decisions with true expert capabilities remains an open challenge.

We propose expert-router coupling loss (ERC loss), a novel auxiliary loss for MoE models that tightly couples routers and experts with negligible overhead. The loss is based on interpreting the router parameter matrix $\boldsymbol{R} \in \mathbb{R}^{n \times d}$ as cluster centers, where each row $\boldsymbol{R}[i]$ serves as the center for the token set $\mathcal{X}_i$ routed to expert $i$. The ERC loss comprises three key steps:

(1) Each $\boldsymbol{R}[i]$ is augmented with bounded random noise $\boldsymbol{\delta}_i$ to obtain $\tilde{\boldsymbol{R}}[i]$, serving as a proxy for tokens in $\mathcal{X}_i$. Here, $\boldsymbol{\delta}_i$ is bounded by half the minimum distance between adjacent cluster centers,

---

*Work done during the internship of Ang Lv (`anglv@ruc.edu.cn`) at Bytedance Seed.

†Corresponding author: Yiyuan Ma (`mayiyuan.unicorn@bytedance.com`).

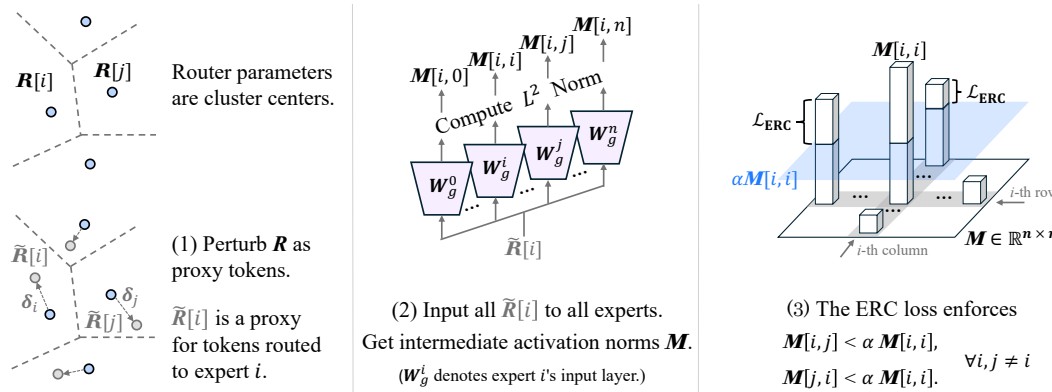

Figure 1: Three steps for computing the expert-router coupling loss.

ensuring that the noise simulates input variations within $\mathcal{X}_i$ while preventing the crossing of cluster boundaries.

(2) Inspired by prior works (Geva et al., 2021; Liu et al., 2023; Lv et al., 2025), the intermediate activation norm serves as an indicator of how well its capabilities align with the token. We measure the intermediate activation norms of all experts that take $\tilde{R}[i]$ as input. This step produces a matrix $M \in \mathbb{R}^{n \times n}$, with $M[i,j]$ being the activation norm from expert $j$ given input $\tilde{R}[i]$.

(3) For all $i \neq j$, the ERC loss imposes a penalty wherever the off-diagonal elements $M[i,j]$ or $M[j,i]$ exceed $\alpha M[i,i]$, where $\alpha$ is a scalar hyperparameter:

$$\mathcal{L}_{\text{ERC}} = \frac{1}{n^2} \sum_{i=1}^{n} \sum_{j \neq i}^{n} \left( \max\left( M[i,j] - \alpha M[i,i], 0 \right) + \max\left( M[j,i] - \alpha M[i,i], 0 \right) \right).$$

Minimizing it tightly couples experts and routers through two effects:

- Expert specialization: The proxy token $\tilde{R}[i]$ elicits the strongest activation from expert $i$ versus all other experts. This indicates that expert $i$ is optimized to best match the features of its assigned token cluster $\mathcal{X}_i$.

- Precise token routing: Expert $i$ is most activated by its designated vector $\tilde{R}[i]$ than to any other $\tilde{R}[j]$ for $j \neq i$. This demonstrates that $R[i]$ aligns well with the capabilities of expert $i$, ensuring that the router assigns to this expert the tokens that need it most.

We conducted large-scale pre-training experiments on models from 3B to 15B parameters, using a total of several trillion tokens. The ERC loss not only significantly enhances model performance and narrows the performance gap with a competitive yet more computationally expensive MoE variant (Lv et al., 2025) but also retains the efficiency of vanilla MoEs.

Furthermore, building on the first effect, we establish that the ERC loss serves as a powerful tool for studying expert specialization. This property arises from two key features of the ERC loss: (1) the specialization level is explicitly controlled by $\alpha$, and (2) the bound of noise $\delta_i$ provides a quantitative measure for this level. Through this lens, we reveal a trade-off between specialization and model performance. Our findings challenge some beliefs about expert specialization that were derived from small-scale experiments. These quantitative and qualitative analysis methods offer new pathways to advance the understanding of MoE models.

## 2 BACKGROUND

**Mixture-of-Experts** Our description follows the prevailing SwiGLU structure used by advanced LLMs (Qwen, 2024; DeepSeek-AI, 2025; OpenAI et al., 2025). An MoE layer consists of $n$ experts, where each expert $i$ is parameterized by three matrices: $W_g^i \in \mathbb{R}^{d \times D}$, $W_p^i \in \mathbb{R}^{d \times D}$, and $W_o^i \in$

$\mathbb{R}^{D \times d}$. The layer also includes a router with the weight matrix $\boldsymbol{R} \in \mathbb{R}^{n \times d}$, which takes a token $\boldsymbol{x} \in \mathbb{R}^d$ as input and outputs an expert weight[1] vector:

$$\boldsymbol{w} = \text{softmax}(\boldsymbol{x}\boldsymbol{R}^{\top}) \in \mathbb{R}^n.$$

Typically, the top-$K$ experts with the highest expert weights are selected to process the token. The processing of $\boldsymbol{x}$ by expert $i$ is given by:

$$E_i(\boldsymbol{x}) = \left(\text{SiLU}(\boldsymbol{x}\boldsymbol{W}_g^i) \odot (\boldsymbol{x}\boldsymbol{W}_p^i)\right) \boldsymbol{W}_o^i,$$

where $\odot$ denotes element-wise multiplication. The final output of the entire MoE layer is the weighted sum of the outputs of the selected experts:

$$\sum_{k}^{K} \boldsymbol{w}[k] E_k(\boldsymbol{x}), \text{ where } k \in \text{Top-K}(\boldsymbol{w}).$$

**Expert-router coupling via denser activation** Autonomy-of-Experts (AoE; Lv et al., 2025) encodes the routing function into expert parameters. AoE factorizes $\boldsymbol{W}_g$ into two $r$-rank matrices $\boldsymbol{W}_{down}^i \in \mathbb{R}^{d \times r}$ and $\boldsymbol{W}_{up}^i \in \mathbb{R}^{r \times D}$. Each expert processes a token up to the point after the $\boldsymbol{W}_{down}^i$ projection. The expert weight vector is computed using the activation norm at this stage:

$$\boldsymbol{w} = \text{softmax}\left(\{\|\boldsymbol{x}\boldsymbol{W}_{down}^i\| \text{ for } i = 1, \ldots, n\}\right).$$

The top-$K$ experts exhibiting the highest activation norms are selected to continue their forward computation, and the others are terminated early. This norm-based selection is justified by the fact that the activation norm of MLPs represents how well their capabilities match their inputs (Geva et al., 2021; Liu et al., 2023). The computational overhead of AoE scales with the number of tokens during both training and inference. Moreover, this inefficiency worsens as the number of experts $n$ increases or the selection count $K$ decreases. Wu et al. (2025) found that using only a small, representative subset of neurons per expert is sufficient for "autonomous expert selection," reducing but not eliminating AoE's token-dependent cost.

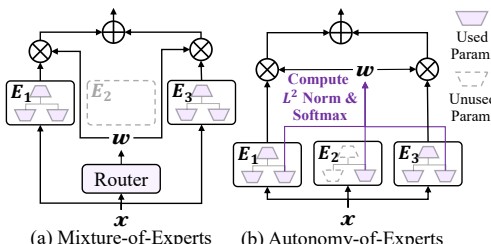

(a) Mixture-of-Experts (b) Autonomy-of-Experts

Figure 2: The overview of MoE and AoE models.

Pham et al. (2024) use experts' *final* output norms to supervise router logits. There is no inference overhead but the model is fully dense-activated during training, contradicting the core sparsity principle of MoE. Therefore, we include it only for background discussion, not as a baseline.

## 3 METHOD

After analyzing the strengths and limitations of prior work, we distill three design principles to ensure a lightweight, effective, and practically applicable enhancement for expert-router coupling in MoE-LLMs:

(1) Routers must be retained in MoE architectures to preserve routing efficiency.

(2) An auxiliary loss that enables interaction between experts and routers can strengthen their coupling.

(3) The loss must have complexity independent of the number of input tokens and must not introduce activation density beyond that of a vanilla MoE.

Below, we introduce expert-router coupling loss, which fulfills all these principles.

---

[1]In this paper, "weight" refers to the relative contribution of each expert's output or the strength of the loss function. Please carefully distinguish between "weight" and "parameter."

### 3.1 Expert-Router Coupling Loss

The expert-router coupling (ERC) loss is motivated by a clustering-based interpretation of MoE routing: The routing mechanism in traditional MoE models can be interpreted as a clustering process, where router parameters $\boldsymbol{R} \in \mathbb{R}^{n \times d}$ are viewed as $n$ cluster centers. For any input token $\boldsymbol{x} \in \mathbb{R}^d$, the router computes an $n$-dimensional logit vector representing the weight assigned to each expert. Specifically, the weight for expert $i$ is derived from the inner product between $\boldsymbol{x}$ and the cluster center $\boldsymbol{R}[i]$. When $\boldsymbol{x}$ belongs to the cluster centered at $\boldsymbol{R}[i]$, this inner product is maximized[2], making expert $i$ the top choice.

A key advantage of this clustering view is that it enables probing an expert's responsiveness to a set of tokens without feeding every token to all experts, unlike prior methods (See §2). Instead, we leverage each cluster center $\boldsymbol{R}[i]$ as a proxy for tokens routed to expert $i$ (denoted as $\mathcal{X}_i$), enabling us to derive intermediate activations and evaluate how well the expert aligns with a proxy token.

Our ERC loss is computed in three key steps:

(1) For each cluster center $\boldsymbol{R}[i]$, we create a perturbed proxy token $\tilde{\boldsymbol{R}}[i] = \boldsymbol{R}[i] \odot \boldsymbol{\delta}_i$. $\boldsymbol{\delta}_i \in \mathbb{R}^d$ is bounded multiplicative random noise, which we elaborate in §3.2. This noise ensures the proxy generalizes to tokens in $\mathcal{X}_i$. *Notably, the perturbed $\tilde{\boldsymbol{R}}$ is used only for loss computation*; routing still uses the clean $\boldsymbol{R}$ to compute router logits, as in standard MoEs.

(2) Each proxy token is processed by the $\boldsymbol{W}_g$ parameter of all $n$ experts, yielding a total of $n^2$ intermediate activations. The $L^2$ norm of each activation is computed to form a matrix $\boldsymbol{M} \in \mathbb{R}^{n \times n}$, where $\boldsymbol{M}[i, j]$ corresponds to the norm from expert $j$ given input $\tilde{\boldsymbol{R}}[i]$:

$$\boldsymbol{M}[i,j] = \left\| \tilde{\boldsymbol{R}}[i] \cdot \boldsymbol{W}_g^j \right\|.$$

(3) To enforce expert-router coupling, for all $i$ and $j \neq i$, the ERC loss imposes two constraints, where a scalar $\alpha \in [0, 1]$ determines their strength:

$$\boldsymbol{M}[i,j] < \alpha \boldsymbol{M}[i,i], \tag{1}$$

$$\boldsymbol{M}[j,i] < \alpha \boldsymbol{M}[i,i]. \tag{2}$$

Constraint 1 ensures the proxy token $\tilde{\boldsymbol{R}}[i]$ activates its corresponding expert $i$ more than any other expert $j$. Since tokens similar to $\boldsymbol{R}[i]$ are routed to expert $i$, and given their similarity to $\tilde{\boldsymbol{R}}[i]$, they also elicit a stronger activation in expert $i$ than in other experts. This strongest activation indicates that expert $i$ is optimized to develop capabilities best suited to $\mathcal{X}_i$ (Lv et al., 2025).

Constraint 2 requires that expert $i$ responds more strongly to its own proxy token $\tilde{\boldsymbol{R}}[i]$ than by any other $\tilde{\boldsymbol{R}}[j]$. This ensures each $\boldsymbol{R}[i]$ accurately represents expert $i$, guaranteeing that tokens most needing expert $i$ are correctly routed to it.

As $\alpha$ decreases, the two constraints become stricter, thereby enforcing stronger expert-router coupling. Additionally, $\alpha$ enables flexible regulation of specialization: a smaller $\alpha$ increases the gap between $\boldsymbol{M}[i,i]$ and $\boldsymbol{M}[i,j]$, reflecting greater expert specialization as experts exhibit more differentiated responses to the same inputs. This feature makes the ERC loss a useful tool for investigating expert specialization and provides deeper insight into MoE behavior, as demonstrated in §4.2.

We translate these two constraints into expert-router coupling loss, formally defined as:

$$\mathcal{L}_{\text{ERC}} = \frac{1}{n^2} \sum_{i=1}^{n} \sum_{j \neq i}^{n} \left( \max\left(\boldsymbol{M}[i,j] - \alpha \boldsymbol{M}[i,i], 0\right) + \max\left(\boldsymbol{M}[j,i] - \alpha \boldsymbol{M}[i,i], 0\right) \right). \tag{3}$$

The three steps for computing expert-router coupling loss are illustrated in Figure 1. For implementation details, we provide PyTorch-style pseudocode in Figure 8.

---

[2]This assumes all $\boldsymbol{R}[i]$s have comparable norms. We confirm that the models used in our experiments adhere to this assumption.

## 3.2 BOUNDED RANDOM NOISE FOR GENERATING PROXY TOKENS

The perturbed proxy token $\tilde{\boldsymbol{R}}[i] = \boldsymbol{R}[i] \odot \boldsymbol{\delta}_i$ makes expert $i$'s coupling generalize effectively from $\boldsymbol{R}[i]$ alone to $\mathcal{X}_i$. To ensure the perturbed point $\tilde{\boldsymbol{R}}[i]$ remains within its original cluster, we require a bounded perturbation. We therefore model the noise $\boldsymbol{\delta}_i$ as a multivariate uniform distribution, $\boldsymbol{\delta}_i \sim \mathcal{U}(1 - \epsilon_i, 1 + \epsilon_i)^d$. Let $j = \arg\min_{j^* \neq i} \|\boldsymbol{R}[i] - \boldsymbol{R}[j^*]\|$ be the nearest cluster center. For the noise level $\epsilon$ to be sufficient to avoid perturbing the cluster, it must satisfy:

$$\epsilon_i \leq \frac{\|\boldsymbol{R}[i] - \boldsymbol{R}[j]\|}{2\|\boldsymbol{R}[i]\|}. \tag{4}$$

The derivation of this bound is provided in Appendix B. We set $\epsilon_i$ to its maximum value, i.e., the right-hand side of this inequality. Notably, the value of $\epsilon_i$ is dynamically computed at each layer and every training step.

## 3.3 EFFICIENCY ANALYSIS

**Training efficiency**  In a standard MoE layer, $T$ tokens are processed by $K$ experts, resulting in a total computational cost of $6TKDd$ FLOPs. expert-router coupling loss introduces only $2n^2Dd$ additional FLOPs, a cost that is negligible in practical pre-training setups where $K$ is often in the millions. In contrast, AoE introduces an additional overhead of $2T(n-K)dr$ FLOPs (recall that $r$ is AoE's factorization rank; see §2). Given that typical MoE-LLMs operate at sparsity levels far below 25% (i.e., $n > 4K$), this overhead ratio exceeds $r/D$, making it prohibitive. A detailed breakdown of the FLOP calculations supporting the above theoretical analysis is provided in Appendix C.1.

The efficiency of our method is confirmed in practice. The ERC loss maintains low overhead during LLM pre-training under multiple parallelism strategies, adding only $0.2$–$0.8\%$ overhead in our experiments. We provide a complete analysis of these real-world distributed conditions and measured throughputs in Appendix C.2.

**Overhead-free inference**  Expert-router coupling loss introduces no additional inference overhead, since it is not applied at that stage. In contrast, AoE maintains the same forward computation, along with its associated overhead.

## 4 EXPERIMENTS

### 4.1 EXPERIMENTAL SETTINGS

We compare the ERC-loss-augmented MoE against both the vanilla MoE and AoE baselines. All models are trained from scratch with 3B parameters. This parameter size is chosen because it represents the largest scale at which we could successfully train the AoE model under our available resources. Our implementation is based on OLMoE (Muennighoff et al., 2025). The models comprise 12 layers with $d = 1536$ and $D = 768$. Each Transformer (Vaswani et al., 2017) layer has 16 attention heads and $n = 64$ experts, where $K = 8$ experts are selected per token. For the AoE model, we set $r = 512$ to ensure consistent total parameter count. The number of activated parameters is 500M. Each model is trained on 500B tokens from the open-source dataset `dolmap-v1.5-sample` (Soldaini et al., 2024), using a batch size of 3 million tokens. We use the AdamW optimizer (Loshchilov & Hutter, 2019) with $(\beta_1, \beta_2) = (0.9, 0.95)$, a weight decay of 0.1, and a learning rate of 4e-4 with a cosine schedule decaying to 4e-5. A load balancing loss (Fedus et al., 2022) with a weight of 0.01 is applied consistently in all experiments.

For simplicity, the loss weight of the ERC loss is fixed at 1, and we use $\alpha = 1$ by default if not specified.

We evaluate LLMs on the following tasks: ARC-Challenge (Clark et al., 2018), CommonsenseQA (Talmor et al., 2019), COPA (Roemmele et al., 2011), BoolQ (Clark et al., 2019), HellaSwag (Zellers et al., 2019), OpenbookQA (Mihaylov et al., 2018), SciQ (Welbl et al., 2017), Social IQa (Sap et al., 2019), WinoGrande (Sakaguchi et al., 2021), and MMLU (Hendrycks et al., 2021a).

## 4.2 PERFORMANCE, EFFICIENCY, AND LOAD BALANCING

Figure 3(a) reports the average accuracy across all tasks and task-specific results are presented in Figure 9. It shows that the ERC-loss-augmented MoE achieves stable performance gains, which significantly outperforms the vanilla MoE and narrows the gap between AoE and MoE.

In terms of efficiency, MoE models with and without ERC loss have nearly identical throughput and memory costs. By contrast, AoE requires **1.6×** more training hours and **1.3×** higher memory usage, limiting further scaling due to impractical training times and out-of-memory issues.

Expert-router coupling loss is compatible with the load balancing loss. As shown in Figure 3(b), the difference in load balancing loss between MoE combined with $\mathcal{L}_{\text{ERC}}$ and the vanilla MoE is on the order of $10^{-5}$. This difference is negligible given that the overall load balancing loss magnitude remains around $10^{-2}$. By comparison, the loss difference between AoE and vanilla MoE is approximately $4 \times 10^{-4}$. Although this difference is still small, it is notably larger than the difference exhibited by ours.

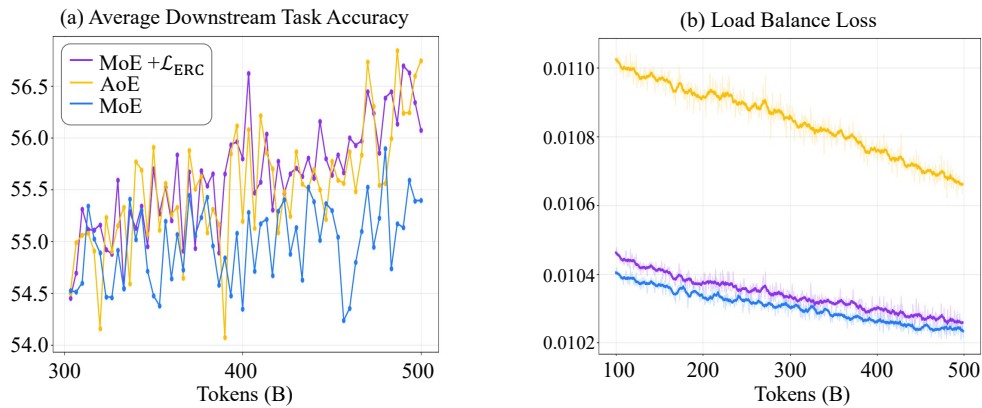

Figure 3: The 3B-scale MoE with ERC loss achieves substantial, stable gains while maintaining effective load balancing. Figure 9 shows task-specific details.

## 4.3 VALIDATING ERC LOSS IN 15B-PARAMETER MOES

We scale models to 15 billion parameters by increasing $n$ to 256 (keeping $K$=8) and doubling the model depth. This configuration results in a total of 15B parameters with approximately 700M activated. Other training hyper-parameters largely follow the setup in Section 4.1. As a large-scale, high-sparsity model, the AoE method failed to train due to being overly costly and is thus omitted from comparison.

Table 1 shows that the benefits of the ERC loss persist across various public benchmarks more challenging than those used for 3B models, including MMLU (Hendrycks et al., 2021a), C-Eval (Huang et al., 2023), MMLU-Pro (Wang et al., 2024b), AGI-Eval (Zhong et al., 2024), BBH (Suzgun et al., 2023), MATH (Hendrycks et al., 2021b), GSM8K (Cobbe et al., 2021), and TriviaQA (Joshi et al., 2017). The consistent performance improvements demonstrate that our method effectively addresses the expert-router decoupling problem even at scale. Throughout this large-scale training, we observed no loss spikes or abnormal gradients.

Table 1: Scaling to 15B parameters: ERC loss improves performance on challenging benchmarks.

|  | MMLU | C-Eval | MMLU-Pro | AGI-Eval | BBH | MATH | GSM8K | TriviaQA |
|---|---|---|---|---|---|---|---|---|
| MoE | 63.2 | 67.5 | 31.0 | 42.0 | 44.3 | 25.7 | 45.2 | 47.2 |
| MoE + $\mathcal{L}_{\text{ERC}}$ | 64.6 | 69.0 | 31.9 | 44.2 | 45.6 | 26.1 | 45.8 | 49.1 |

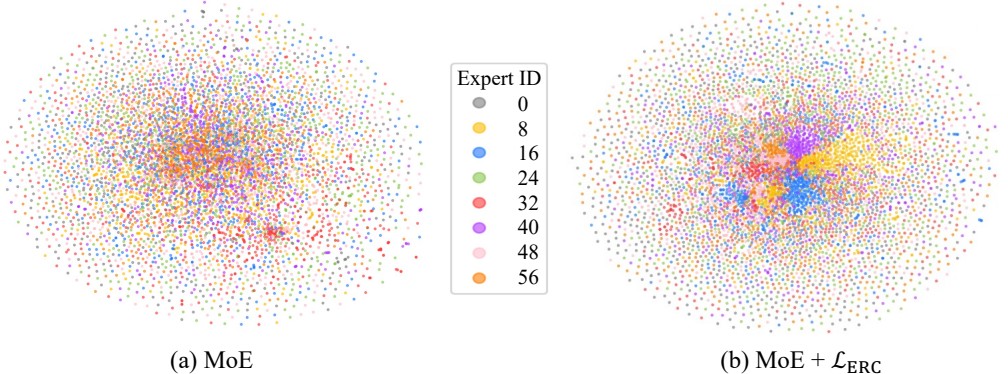

(a) MoE        (b) MoE + $\mathcal{L}_{\text{ERC}}$

Figure 4: t-SNE projections of $\boldsymbol{W}_g$ in MoE experts trained without and with the ERC loss. Our ERC loss provides greater expert specialization.

### 4.4 THE ERC LOSS IS AN EFFECTIVE TOOL FOR EXPLORING EXPERT SPECIALIZATION

With the ERC loss, experts are more specialized, as they exhibit greater discrimination between tokens they process and those they do not, compared to vanilla MoE (without the ERC loss). An intuitive demonstration of this specialization comes from visualizing expert parameters. Following (Yang et al., 2025), we use t-SNE (van der Maaten & Hinton, 2008) to project each row of $\boldsymbol{W}_g^i$ (where $i \mod 8 = 0$) from layer 6 (the middle depth) onto a 2D point. As shown in Figure 4, experts in vanilla MoE lack specialization, as their parameter features do not form meaningful clusters. By contrast, MoE enhanced with the ERC loss exhibits more distinct clusters, indicating specialized capabilities.

Beyond merely promoting specialization, the ERC loss can also serve as a powerful tool for exploring it. We show this capability through two features below.

**Feature 1: $\alpha$ enables a controllable investigation into optimal specialization** In the ERC loss, $\alpha$ governs the coupling strength between experts and the router. When $\alpha = 0$, the ERC loss encourages $\boldsymbol{R}[i]$ to be orthogonal to the parameters of other experts, thereby maximizing specialization. Conversely, when $\alpha \to 1$, the loss permits smaller differences in how all experts' responsiveness to $\boldsymbol{R}[i]$, thus reducing specialization. Notably, $\alpha = 1$ only weakens the ERC loss's constraints to their maximum extent; it still retains a degree of specialization stronger than the spontaneously emerged specialization in a vanilla MoE model.

**Feature 2: $\epsilon$ provides a quantitative measure for specialization** The noise level $\epsilon$ exhibits a strong correlation with $\alpha$, and it can reflect changes in expert specialization throughout the training process. This correlation exists because as $\alpha$ increases, experts are allowed to be more homogeneous. This growing homogeneity among experts, in turn, reduces the separation between the cluster centers in the router as they are tightly coupled. A smaller separation between cluster centers ultimately derives a smaller $\epsilon$. Thus, $\epsilon$ is a quantitative metric tracking expert specialization.

**Experiments** The following experiments support these two features. In Figure 5(a), we plot $\epsilon$ at each training step across a parameter search over $\alpha \in \{0.4, 0.6, 0.8, 1.0\}$. Consistent with our analysis, increasing $\alpha$, which reduces expert specialization, indeed leads to a corresponding decrease in $\epsilon$. Note that measuring router cluster distance is uninformative in vanilla MoE training without the ERC loss, because the router and experts are uncoupled and cluster distances do not reflect expert capability dynamics. We further compared downstream task performance across different values of $\alpha$. Figure 5(b) shows that all tested $\alpha$ values outperform the vanilla MoE model. This not only confirms the robust effectiveness of the ERC loss but also demonstrates that the specialization spontaneously formed by vanilla MoE models is inadequate.

**The optimal specialization degree** Figure 5(b) shows that pursuing extreme specialization is not advisable, as model performance degrades with overly strict $\alpha$. This highlights a trade-off between

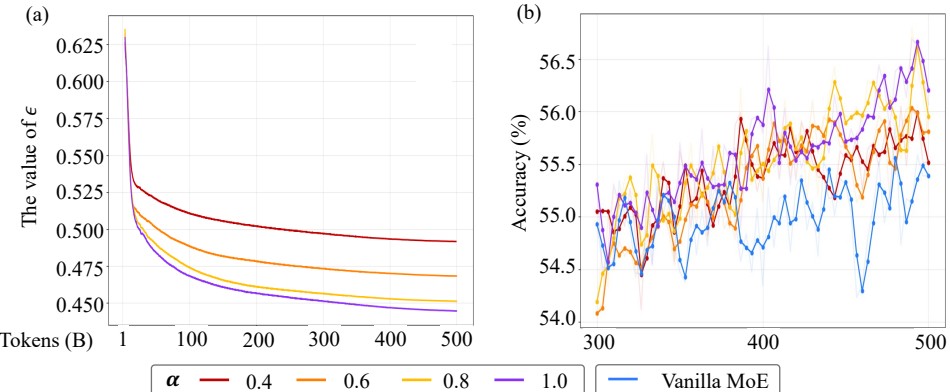

Figure 5: (a) Since routers are deeply coupled with experts, the distance between neighboring cluster centers (i.e., the maximum noise level $\epsilon$) quantitatively reflects changes in expert specialization during training, which is controlled by $\alpha$. (b) Downstream performance across different values of $\alpha$.

promoting expert specialization and maintaining effective collaboration, which is under-discussed in previous work.

The optimal specialization degree is influenced by several factors. The core consideration is whether, among all $\binom{n}{K}$ possible expert combinations, an effective $K$-expert set can be assembled for any given input. In general, smaller values of $n$ favor more generalist experts, while larger $n$ can support a higher degree of specialization. However, we currently lack quantitative metrics to characterize "large" or "small" $n$ and $K$ across different models; as a result, determining the optimal trade-off remains largely empirical. For example, in our experiments with a fixed $K = 8$: When $n = 64$, the optimal $\alpha = 1$, suggesting $n = 64$ is not "large" for *our* 3B-parameter models. In contrast, with $n = 256$, we searched for an optimal $\alpha = 0.5$, indicating $n = 256$ is "large" for *our* 15B-parameter models. This trade-off is also shaped by other architectural choices, such as the use of shared experts[3]. A deeper investigation into these interacting factors, reliable quantitative metrics for specialization, and an automated evaluation of the optimal specialization degree for a given model are left as important problems for future works. For practitioners implementing the ERC loss, we recommend starting with $\alpha = 1$, which eliminates expert decoupling and should provide some gains. Further improvement may be achieved by experimenting with lower $\alpha$ values, depending on the specific configuration of your model.

Several studies (Guo et al., 2026; Liu et al., 2024; Hendawy et al., 2024) have promoted specialization via expert output orthogonality. We argue, however, that orthogonalizing expert outputs does not equate to achieving *extreme* specialization, as the magnitude (norm) of an expert's response to a token remains unconstrained. Moreover, finding a set of orthogonalized high-dimensional vectors is not difficult, making it unclear whether such orthogonality yields sufficiently discriminative representations. Consequently, one should not interpret these fine-tuning experiments as supporting a broad claim that "more specialization is always better." On a separate note, orthogonality among router embeddings (Baidu-ERNIE-Team, 2025) is only weakly correlated with specialization, since the router and experts are typically decoupled. As demonstrated in ablation studies, enforcing router orthogonality might not be a critical factor for pre-training MoE models.

## 4.5 ABLATION STUDIES

**Choice of activations for computing $M$**  We considered five candidates for calculating $M$: using the norms of (a) $\tilde{R}W_g$, (b) $\tilde{R}W_p$, (c) SiLU($\tilde{R}W_g$), (d) the post-SwiGLU activations (i.e., SiLU($\tilde{R}W_g$) $\odot$ $\tilde{R}W_p$), and (e) experts' final outputs (i.e., (SiLU($\tilde{R}W_g$) $\odot$ $\tilde{R}W_p$)$W_o$). As shown in Figure 6(a), $\tilde{R}W_g$ is the most effective among all alternatives. While using the final output

---

[3]A shared expert satisfies general input requirements, allowing the remaining experts to be more specialized even with the same $n$.

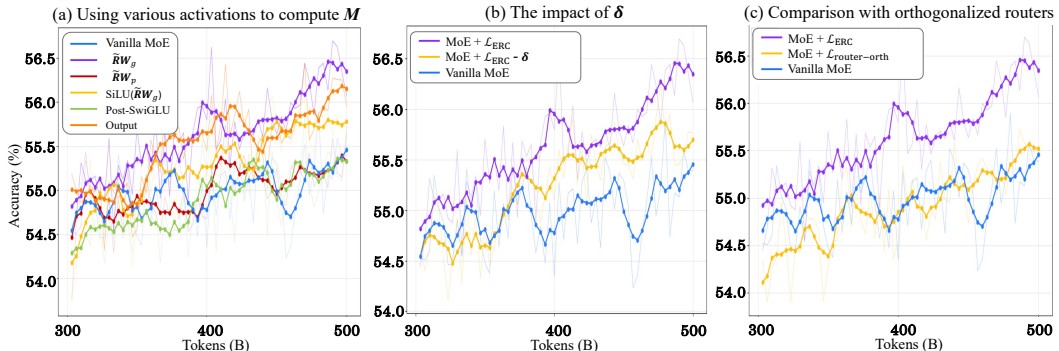

Figure 6: Results of ablation studies. For detailed task-specific results, please refer to Figure 9.

achieves comparable performance, it incurs a higher cost. We therefore adopt $\tilde{R}W_g$ as our default choice.

**Random noise $\delta$ enables the generalization of coupling** The random noise $\delta$ allows $\tilde{R}[i]$ to better capture the samples within $\mathcal{X}_i$. To validate its importance, we conducted an ablation study where we trained an MoE with the ERC loss but removed $\delta$. Specifically, we computed $M$ directly using the original $R$ instead of the noise-augmented $\tilde{R}$. As shown in Figure 6(b), removing $\delta$ greatly degrades performance. This is because the coupling between routers and experts becomes overfitted to $R$, failing to generalize to the real inputs that $R[i]$s represent.

**Comparison with contrastive regularization solely on routers** The router orthogonalization loss (Baidu-ERNIE-Team, 2025) requires $\hat{R}$ (the row-wise normalization of $R$) to satisfy:

$$\hat{R}\hat{R}^{\top} = I.$$

As shown in Figure 6(c), the orthogonalization loss yields only limited gains. We attribute this to our finding that the router embeddings in our baseline MoE model are already nearly orthogonal, with an average absolute cosine similarity of 0.15. This value corresponds to angles between router embeddings mostly ranging from arccos(0.15) = 81° to arccos(-0.15) = 99°. Notably, we do not imply that all MoEs always have nearly orthogonal router embeddings, as this may depend on the data or specific architecture; we report this only as a characteristic of our models, which explains the limited gains from the orthogonalization loss.

This result further demonstrates that weak coupling between routers and experts is a more critical issue than imperfect orthogonality in router embeddings. The significant gains from ERC, even when applied to a baseline with already near-orthogonal routers, provide clear evidence.

Furthermore, it is important to note that even if both routers and experts are orthogonalized, there is no guarantee that each $R[i]$ will be aligned with $W_g^i$. Therefore, the ERC loss cannot be reduced to contrastive techniques applied individually to routers or experts, such as orthogonalization loss.

**Additional analyses** Appendix A provides analyses addressing several frequently asked questions, including the effect of $\alpha > 1$, and verifying that the model decreases the ERC loss by learning meaningful coupling rather than by manipulating parameter norms.

## 5 RELATED WORKS

**Auxiliary loss for MoEs** Auxiliary losses are crucial for training large-scale MoE models. Most existing work in this area focuses primarily on enhancing training stability. For instance, many studies have proposed auxiliary losses to address load balancing challenges (Fedus et al., 2022; Qiu et al., 2025; Wang et al., 2024a); Zoph et al. (2022) introduced the z-loss, which penalizes excessively large logits in the gating network to enable stable training. MoE concepts have also inspired mixtures of attention heads or entire layers (Gong et al., 2024; Lin et al., 2024), where

auxiliary losses play a critical role in effective optimization. Our ERC loss is the first tailored to strengthen the expert-router coupling. Other related auxiliary losses enhancing expert specialization or orthogonality are discussed below.

**Expert specialization**     Dai et al. (2024) introduced a shared expert to handle general capabilities, encouraging the others to be more specialized. Guo et al. (2026) proposes an auxiliary loss to minimize the pairwise projections of the selected top-$K$ experts' outputs for each token, reducing expert overlap but incurring high cost due to $K^2$ cosine similarity calculations per token. Other methods scale the number of tiny experts to millions, making each expert more atomic and thus more specialized (Yang et al., 2025; Park et al., 2025; He, 2024), but are memory-bounded. Beyond efficiency, these methods face three major limitations: (1) no quantitative control over specialization degree; (2) no exploration of the specialized-generalized ability trade-off; and (3) failure to strengthen expert-router coupling. Our method addresses all three, both efficiently and effectively.

Some works (Guo et al., 2026; Liu et al., 2024; Hendawy et al., 2024) maximize specialization by training orthogonal experts, but their evaluations are based on fine-tuning (or reinforcement learning) experiments. We contend that orthogonalizing expert outputs is not equivalent to achieving extreme specialization, and further, that the optimal degree of specialization is a complex problem affected by various factors and requires further exploration.

**Contrastive learning**     Constraints 1 and 2 bear similarity to contrastive learning (Chen et al., 2020; van den Oord et al., 2019; Khosla et al., 2020). Some MoE research (Luo et al., 2024; Guo et al., 2026) applied contrastive learning to expert outputs, encouraging specialization. However, naively applying contrastive learning to either routers or experts leaves the weak expert-router coupling unaddressed.

## 6 CONCLUSIONS

The weak coupling between router decisions and expert capabilities limits MoE models in multiple important aspects. We propose expert-router coupling loss that tightly couples router parameters with their corresponding experts. The proposed ERC loss improves MoE-based LLMs on downstream tasks while incurring negligible training overhead. In addition, it exhibits several desirable properties that not only provide deeper insight into the behavior of MoE models but also offer a promising tool for future research on expert specialization.

## STATEMENTS ON ETHICS, REPRODUCIBILITY, AND LLM USAGE

Our research does not raise ethical issues. For reproducibility, we used public data and code, and provide algorithm code in Figure 8. We used LLMs solely for typo checking.

### ACKNOWLEDGMENTS

We thank Songhao Wu and Ziteng Wang for their insightful discussions. We are grateful to Ruobing Xie, Yining Qian, and Kaiyi Zhang for proofreading and writing suggestions. Additionally, we sincerely acknowledge the anonymous ICLR reviewers for their constructive comments and questions, which have greatly improved this work.

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

# A  FREQUENTLY ASKED QUESTIONS

## A.1  WHAT HAPPENS IF $\alpha > 1$?

Some readers might be interested in the value of $\alpha$ at which the ERC loss degenerates to no effective constraints, and the trained model consequently degenerates to a vanilla MoE. For our baseline MoE, we seek the minimum $\alpha$ that zeros the ERC loss computed from the $\mathbf{M}$ matrices of the last checkpoint. Table 2 shows that achieving zero ERC loss across all layers requires $\alpha = 5$ in our pre-trained vanilla MoE baseline. This provides direct evidence that the router-expert coupling in the vanilla MoE is very weak.

We further pre-trained 3B MoE models with the ERC loss at $\alpha$ values of 2 and 3. It is important to note that this experiment is to only demonstrate the effects of loosening the ERC constraints. We do not recommend using $\alpha > 1$ in practice, as it contradicts our core motivation: the router and experts will shift from a state of no mismatch toward looser coupling constraints, ultimately causing the model to degenerate into a vanilla MoE. As shown in Figure 7, the model with $\alpha = 2$ yields only limited improvement, while the model with $\alpha = 3$ shows almost no improvement over the vanilla MoE.

## A.2  DO MODELS REDUCE ERC LOSS THROUGH MANIPULATING PARAMETER NORMS?

Some readers might assume that simply increasing or decreasing the norms of certain router embeddings or experts will increase the diagonal entries of $\mathbf{M}$, thereby reducing the ERC loss. Below, we (1) explain that any attempt to reduce one term of the ERC loss by manipulating norms will simultaneously increase other terms, and (2) present detailed parameter norms as evidence.

Note that $\mathbf{M}[i,j]$ can be written as $\|\mathbf{R}[i]\|\|\mathbf{W}_g^j\|\tilde{\mathrm{cos}}_{i,j}$, where $\tilde{\mathrm{cos}}_{i,j}$ denotes the averaged cosine similarity between $\mathbf{R}[i]$ and $\mathbf{W}_g^j$.[4]

Increasing $\|\mathbf{R}[i]\|$ decreases the following loss term (as the second term increases):

$$\|\mathbf{R}[j]\|\|\mathbf{W}_g^i\|\tilde{\mathrm{cos}}_{j,i} - \|\mathbf{R}[i]\|\|\mathbf{W}_g^i\|\tilde{\mathrm{cos}}_{i,i}.$$

---

[4]Formally, $\tilde{\mathrm{cos}}_{i,j} = \sum_{k \leq D} \frac{\|\mathbf{W}_g^j[:,k]\|}{\|\mathbf{W}_g^j\|} \cos\theta_{i,j,k}$, where $\theta_{i,j,k}$ is the angle between $\mathbf{R}[i]$ and the $k$-th column of $\mathbf{W}_g^j$. For clarity, we use this compact form.

Table 2: Post-hoc ERC loss evaluation of the vanilla MoE across $\alpha$ values. For clarity, loss values are computed using the original $\boldsymbol{R}$ rather than $\tilde{\boldsymbol{R}}$, making the results deterministic.

| Layer | Value of $\alpha$ | | | | |
|---|---|---|---|---|---|
| | 1 | 2 | 3 | 4 | 5 |
| 0 | 0.87 | 0.69 | 0.26 | 0.00 | 0.00 |
| 1 | 0.42 | 0.28 | 0.10 | 0.00 | 0.00 |
| 2 | 0.45 | 0.19 | 0.00 | 0.00 | 0.00 |
| 3 | 0.25 | 0.15 | 0.00 | 0.00 | 0.00 |
| 4 | 0.28 | 0.08 | 0.00 | 0.00 | 0.00 |
| 5 | 0.24 | 0.22 | 0.00 | 0.00 | 0.00 |
| 6 | 0.22 | 0.15 | 0.00 | 0.00 | 0.00 |
| 7 | 0.21 | 0.13 | 0.00 | 0.00 | 0.00 |
| 8 | 0.15 | 0.05 | 0.00 | 0.00 | 0.00 |
| 9 | 0.16 | 0.00 | 0.00 | 0.00 | 0.00 |
| 10 | 0.21 | 0.09 | 0.00 | 0.00 | 0.00 |
| 11 | 0.50 | 0.44 | 0.20 | 0.20 | 0.00 |

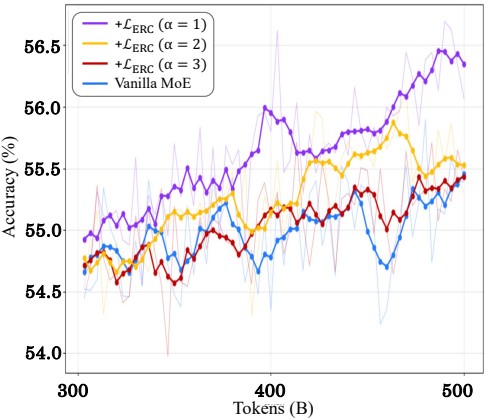

Figure 7: $\alpha > 1$ leads to degeneration into vanilla MoEs

However, it simultaneously increases the following term (as the first term grows):

$$\|\boldsymbol{R}[i]\|\|\boldsymbol{W}_g^j\|\tilde{\cos}_{i,j} - \|\boldsymbol{R}[j]\|\|\boldsymbol{W}_g^j\|\tilde{\cos}_{j,j}.$$

Similarly, any attempt to manipulate the norms of $\boldsymbol{W}_g$ to reduce one part of the loss necessarily increases others, assuming that $\|\boldsymbol{W}_g^j[:,k]\|/\|\boldsymbol{W}_g^j\|$ remains fixed.

This property ensures that the overall ERC loss is minimized only when the router embedding norms are kept comparable and a meaningful coupling is established between routers and their experts.

As shown in the first four columns of Table 3, the average parameter norms for models trained with and without the ERC loss are comparable. Meanwhile, the lower standard deviation under the ERC loss reflects more consistent norms across both router embeddings and experts. In the last two columns of the table, we present the ERC loss for each model. The ERC loss is significantly higher in the baseline model despite its similar average parameter norms.

## B    DETERMINING THE MAXIMUM MULTIPLICATIVE NOISE LEVEL

In what follows, we write $\boldsymbol{R}_i$ for $\boldsymbol{R}[i]$ and $\boldsymbol{R}_{i,k}$ for the $k$-th element of $\boldsymbol{R}[i]$ to avoid excessive brackets. $\boldsymbol{\delta}_i$ is a random vector where each component $\boldsymbol{\delta}_{i,k}$ follows a uniform distribution $\mathcal{U}(1 - \epsilon, 1 + \epsilon)$, and all components are mutually independent. The perturbed point is given by:

$$\tilde{\boldsymbol{R}}_i = (\boldsymbol{\delta}_{i,1}(\boldsymbol{R}_{i,1}), \boldsymbol{\delta}_{i,2}(\boldsymbol{R}_{i,2}), \ldots, \boldsymbol{\delta}_{i,d}(\boldsymbol{R}_{i,d})).$$

To ensure that $\tilde{\boldsymbol{R}}_i$ remains in the same cluster as $\boldsymbol{R}_i$, it must satisfy:

$$\|\tilde{\boldsymbol{R}}_i - \boldsymbol{R}_i\|^2 < \|\tilde{\boldsymbol{R}}_i - \boldsymbol{R}_j\|^2,$$

where $j = \arg\min_{j^* \neq i} \|\boldsymbol{R}[i] - \boldsymbol{R}[j]\|$.

Expanding the squared norms on both sides of the inequality yields:

$$\|\tilde{\boldsymbol{R}}_i - \boldsymbol{R}_i\|^2 = \sum_{k=1}^{d} (\boldsymbol{\delta}_{i,k} - 1)^2 (\boldsymbol{R}_{i,k})^2,$$

Table 3: The first four columns show parameter norms for models trained with and without ERC loss, while the last two show the corresponding layer-wise ERC loss. These results show that MoE + $\mathcal{L}_{\text{ERC}}$ learns a meaningful coupling, rather than trivially minimizing the loss through norm manipulation. All values are evaluated on the last checkpoint.

| Layer | $\|\boldsymbol{R}[i]\|$ | | $\|\boldsymbol{W}_g^i\|$ | | $\mathcal{L}_{\text{ERC}}$ Values | |
| | Baseline | $+\mathcal{L}_{\text{ERC}}$ | Baseline | $+\mathcal{L}_{\text{ERC}}$ | Baseline | $+\mathcal{L}_{\text{ERC}}$ |
|---|---|---|---|---|---|---|
| 0 | 1.85±0.39 | 1.67±0.31 | 25.46±3.93 | 24.14±3.02 | 0.87 | 0.00 |
| 1 | 1.25±0.13 | 1.13±0.12 | 30.14±0.68 | 29.42±0.69 | 0.42 | 0.00 |
| 2 | 1.17±0.12 | 1.07±0.09 | 30.63±0.77 | 29.88±0.76 | 0.45 | 0.00 |
| 3 | 1.10±0.08 | 1.01±0.07 | 30.18±0.77 | 29.42±0.78 | 0.25 | 0.00 |
| 4 | 1.03±0.08 | 0.89±0.05 | 30.59±1.21 | 29.88±1.09 | 0.28 | 0.00 |
| 5 | 0.93±0.08 | 0.87±0.06 | 30.33±1.13 | 29.86±1.06 | 0.24 | 0.00 |
| 6 | 0.86±0.08 | 0.83±0.07 | 30.65±1.15 | 29.82±1.11 | 0.22 | 0.00 |
| 7 | 0.82±0.07 | 0.75±0.06 | 30.56±1.20 | 29.96±1.16 | 0.21 | 0.00 |
| 8 | 0.77±0.06 | 0.76±0.06 | 30.46±1.02 | 29.82±0.88 | 0.15 | 0.00 |
| 9 | 0.80±0.07 | 0.74±0.06 | 30.58±0.88 | 29.86±0.79 | 0.16 | 0.00 |
| 10 | 0.74±0.08 | 0.69±0.06 | 30.80±1.03 | 30.16±0.89 | 0.21 | 0.00 |
| 11 | 0.80±0.14 | 0.73±0.10 | 32.03±1.46 | 31.50±1.26 | 0.50 | 0.00 |

$$\|\tilde{\boldsymbol{R}}_i - \boldsymbol{R}_j\|^2 = \sum_{k=1}^{d}(\boldsymbol{\delta}_{i,k}\boldsymbol{R}_{i,k} - \boldsymbol{R}_{j,k})^2.$$

Substituting into the inequality and simplifying gives:

$$\sum_{k=1}^{d}[2\boldsymbol{\delta}_{i,k}(\boldsymbol{R}_{i,k}(\boldsymbol{R}_{j,k} - \boldsymbol{R}_{i,k}) + (\boldsymbol{R}_{i,k}^2 - \boldsymbol{R}_{j,k}^2)] < 0.$$

To ensure this inequality holds for all realizations of $\boldsymbol{\delta}_i$, we consider the worst-case scenario that maximizes the left-hand side. Define:

$$A_k = 2\boldsymbol{R}_{i,k}(\boldsymbol{R}_{j,k} - \boldsymbol{R}_{i,k}), \quad B = \sum_{k=1}^{d}(\boldsymbol{R}_{i,k}^2 - \boldsymbol{R}_{j,k}^2),$$

so the inequality becomes:

$$\sum_{k=1}^{d} A_k\boldsymbol{\delta}_{i,k} + B < 0. \tag{5}$$

The worst-case $\boldsymbol{\delta}_{i,k}$ is chosen to maximize $\sum A_k\boldsymbol{\delta}_{i,k}$:

$$\boldsymbol{\delta}_{i,k} = \begin{cases} 1 + \epsilon & \text{if } A_k > 0, \\ 1 - \epsilon & \text{if } A_k < 0. \end{cases}$$

Substituting these values gives:

$$\sum_{k=1}^{d} A_k + \epsilon \sum_{k=1}^{d} |A_k| + B < 0. \tag{6}$$

Now simplify $\sum A_k + B$:

$$
\begin{aligned}
\sum A_k + B &= 2\sum \boldsymbol{R}_{i,k}\boldsymbol{R}_{j,k} - 2\sum \boldsymbol{R}_{i,k}^2 + \sum \boldsymbol{R}_{i,k}^2 - \sum \boldsymbol{R}_{j,k}^2 \\
&= 2\sum \boldsymbol{R}_{i,k}\boldsymbol{R}_{j,k} - \sum \boldsymbol{R}_{i,k}^2 - \sum \boldsymbol{R}_{j,k}^2 \\
&= -\left(\sum \boldsymbol{R}_{i,k}^2 - 2\sum \boldsymbol{R}_{i,k}\boldsymbol{R}_{j,k} + \sum \boldsymbol{R}_{j,k}^2\right) \\
&= -\|\boldsymbol{R}_i - \boldsymbol{R}_j\|^2
\end{aligned}
\tag{7}
$$

Substituting equation 7 into equation 6 yields:

$$
-\|\boldsymbol{R}_i - \boldsymbol{R}_j\|^2 + 2\epsilon \sum_{k=1}^{d} |\boldsymbol{R}_{i,k}(\boldsymbol{R}_{j,k} - \boldsymbol{R}_{i,k})| < 0.
$$

Solving for $\epsilon$ gives:

$$
\epsilon_{\max} < \frac{\|\boldsymbol{R}_j - \boldsymbol{R}_i\|^2}{2\sum_{k=1}^{d} |\boldsymbol{R}_{i,k}(\boldsymbol{R}_{j,k} - \boldsymbol{R}_{i,k})|}.
$$

However, computing the denominator of this expression is relatively complex. To balance the efficiency of loss calculation, we instead adopt a tighter upper bound for $\epsilon$.

By the Cauchy-Schwarz Inequality, the following relationship holds:

$$
\sum_{k=1}^{d} |\boldsymbol{R}_{i,k}(\boldsymbol{R}_{j,k} - \boldsymbol{R}_{i,k})| \leq \|\boldsymbol{R}_i\|\|\boldsymbol{R}_j - \boldsymbol{R}_i\|.
$$

Thus, we have:

$$
\epsilon_{\max} = \frac{\|\boldsymbol{R}_j - \boldsymbol{R}_i\|^2}{2\sum_{k=1}^{d} |\boldsymbol{R}_{i,k}(\boldsymbol{R}_{j,k} - \boldsymbol{R}_{i,k})|} \geq \frac{\|\boldsymbol{R}_j - \boldsymbol{R}_i\|^2}{2\|\boldsymbol{R}_i\|\|\boldsymbol{R}_j - \boldsymbol{R}_i\|} = \frac{\|\boldsymbol{R}_j - \boldsymbol{R}_i\|}{2\|\boldsymbol{R}_i\|}.
$$

The term on the right-hand side of the final inequality is the value of $\epsilon$ we used in the main text. This choice ensures that the perturbed $\tilde{\boldsymbol{R}}[i]$ always remains closer in Euclidean distance to $\boldsymbol{R}[i]$ than to any other $\boldsymbol{R}[j \neq i]$.

## C  EFFICIENCY ANALYSIS

Appendix C.1 analyzes the ideal FLOPs cost breakdown of the vanilla MoE, as well as the overhead introduced by AoE and ERC loss. Appendix C.2 discusses efficiency with consideration of the multiple parallelism strategies used in real-world MoE pre-training. Both analyses demonstrate the practicality of our method.

### C.1  FLOPs COST BREAKDOWN OF THREE METHODS

**MoE forward**     Each expert in a MoE layer involves the following operations, with their respective FLOP counts:

- Two matrix multiplications of dimension $T \times d$ with $d \times D$, accounting for $4TdD$ FLOPs. These correspond to the linear transformations parameterized by $\boldsymbol{W}_g$ and $\boldsymbol{W}_p$.

- One element-wise multiplication of $T \times D$ tensors and one SiLU activation applied to a $T \times D$ tensor. The computational cost of these operations is negligible compared to the matrix multiplications.

- One matrix multiplication of dimension $T \times D$ with $D \times d$, contributing $2TDd$ FLOPs. This corresponds to the linear transformation parameterized by $\boldsymbol{W}_o$.

Summing these components gives a total of $6TdD$ FLOPs per expert. For $K$ experts processing $T$ tokens, the total computational cost is therefore $6KTdD$ FLOPs.

**Computational overhead of AoE**  AoE factorizes the expert matrix $\boldsymbol{W}_g \in \mathbb{R}^{D \times d}$ into two low-rank matrices of rank $r$. To maintain the same number of parameters as the original matrix, we require $dr + Dr = Dd$, which gives $r = \frac{Dd}{d+D}$.

The change in FLOPs compared to an MoE is:

$$
T \left( \underbrace{2ndr}_{\text{All experts use } \boldsymbol{W}_{\text{down}}} + \underbrace{2KDr}_{\text{Top-}K \text{ experts use } \boldsymbol{W}_{\text{up}}} - \underbrace{2KDd}_{\text{Top-}K \text{ experts use original } \boldsymbol{W}_g} \right),
$$

where $T$ is the number of tokens. Substituting the value of $r$ and simplifying leads to an extra computational cost of:

$$2T(n - K)dr.$$

**Computational overhead of expert-router coupling loss**  It introduces $n^2$ matrix multiplications, each operating on tensors of shape $1 \times d$ and $d \times D$. In total, this results in $2n^2 Dd$ extra FLOPs.

## C.2 Throughputs under multiple parallelism strategies

We now assess the overhead of the ERC loss in a realistic large-scale pre-training setup that employs both data parallelism (DP) and expert parallelism (EP). As derived in our previous analysis, the computational cost of the ERC loss is equivalent to a forward pass on $n^2/3$ tokens. When distributed across devices, the costs are:

- Base MoE forward: $K \cdot T \,/\, \texttt{dp\_size}$
- ERC overhead: $n \cdot (n \,/\, \texttt{ep\_size}) \,/\, 3$

Consider training our 15B-parameter model with the configuration: $K = 8$, $T = 3 \times 10^6$, $n = 256$, $\texttt{dp\_size} = 64$, and $\texttt{ep\_size} = 8$. In this scenario, the ERC overhead constitutes a mere 0.72% of the base model's forward pass cost. This theoretical estimate is consistent with our empirical measurements: we observe a throughput of 62.03B tokens/day for the baseline versus 61.52B tokens/day for our model, representing only a 0.82% reduction. With a smaller $n = 64$, as in our 3B models trained with $\texttt{dp\_size}=32$ and $\texttt{ep\_size}=1$ (i.e., EP disabled), the overhead ratio drops further to 0.18%. This analysis confirms the practical efficiency of our method.

```python
1   import torch
2   import torch.nn as nn
3   import PseudoExpertClass
4
5   class MoE(nn.Module):
6
7       def __init__(self, args):
8           super().__init__()
9
10          self.experts = PseudoExpertClass(args)
11          # Shape of experts.Wg: (n, D, d)
12          self.R = torch.nn.Parameter(torch.empty(
13              args.n, args.d))
14
15          self.alpha = args.alpha
16
17      def erc_loss(self, M):
18          row_diff = (M - self.alpha * torch.diag(M).unsqueeze(1))
19          row_diff_clamped = torch.clamp(row_diff, min=0.0)
20
21          col_diff = (M - self.alpha * torch.diag(M).unsqueeze(0))
22          col_diff_clamped = torch.clamp(col_diff, min=0.0)
23
24          mask = torch.ones_like(M) - torch.eye(M.size(0), device=M.device)
25          total_diff = (row_diff_clamped + col_diff_clamped) * mask
26
27          return total_diff.mean()
28
29      def get_noisy_router(self, R):
30          with torch.no_grad():
31              norm_R = torch.norm(R, dim=1)
32              distances = torch.cdist(R, R, p=2)
33              distances.fill_diagonal_(float('inf'))
34              min_dist, _ = torch.min(distances, dim=1)
35              eps = min_dist / 2 / norm_R
36
37              low = (1 - eps).unsqueeze(1)
38              high = (1 + eps).unsqueeze(1)
39              noise = torch.rand_like(R)
40          return (low + noise * (high - low)) * R
41
42      def forward(self, x):
43
44          erc_loss = 0.0
45          if self.training:
46              R = self.get_noisy_router(self.R)
47              M = torch.norm(torch.einsum('jDd,id->ijD', self.experts.Wg,
                    R), dim=-1)
48              erc_loss = self.erc_loss(M)
49
50          logits = x.view(-1, x.shape[-1]) @ self.R.T
51          scores = logits.softmax(dim=-1)
52          expert_weights, expert_indices = torch.topk(scores, dim=-1)
53
54          return self.experts(x, expert_weights, expert_indices), erc_loss
```

Figure 8: Pseudo code for expert-router coupling loss in PyTorch.

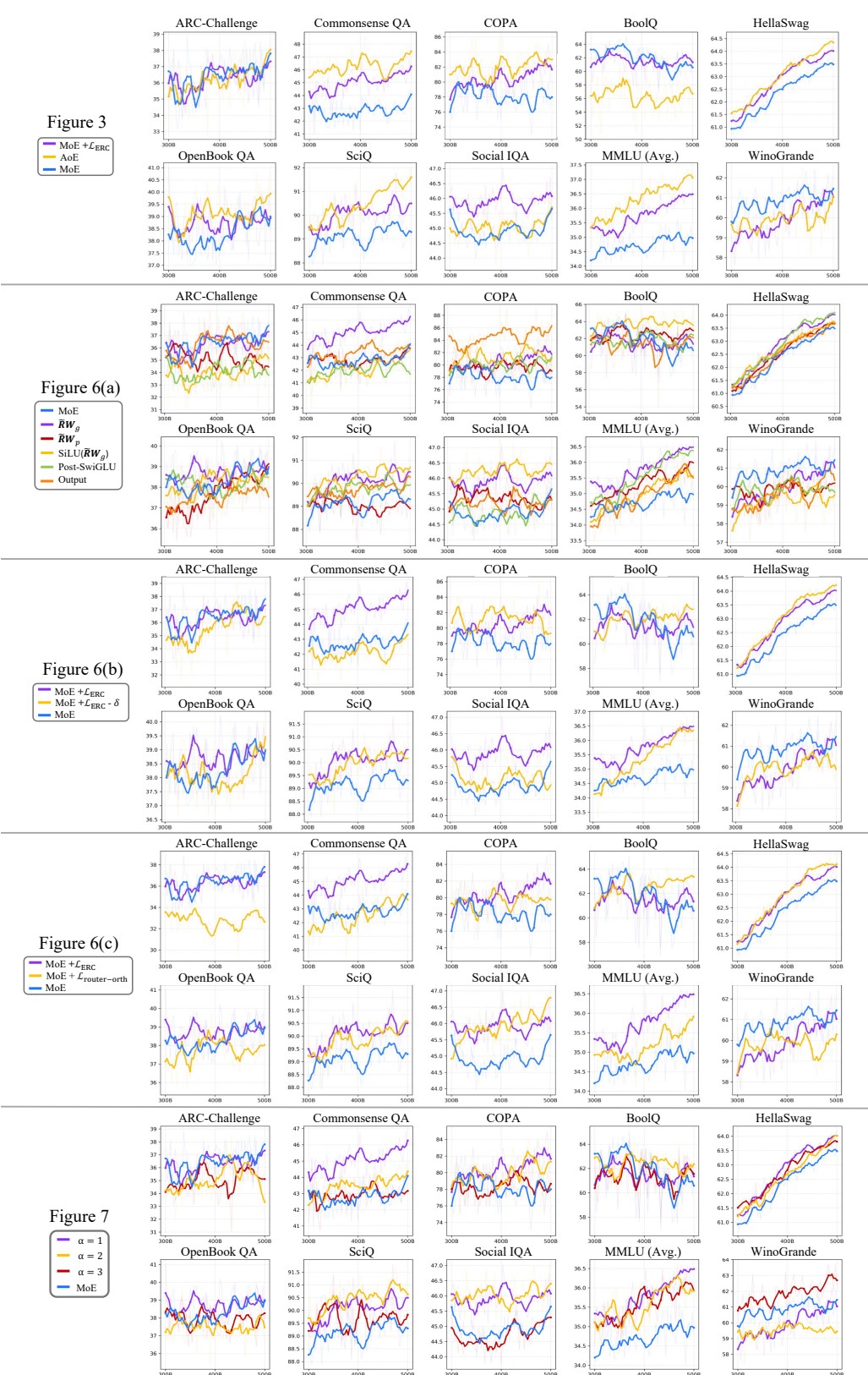

Figure 9: Task-specific downstream results for previous experiments.

