# OpenReview forum: "Coupling Experts and Routers in Mixture-of-Experts via an Auxiliary Loss"
_ICLR.cc/2026/Conference — ICLR 2026 Oral_

### Official Review · Reviewer_WsdJ · 2025-10-31

**Soundness:** 2
**Presentation:** 3
**Contribution:** 2
**Rating:** 6
**Confidence:** 3

**Summary:**

This paper proposes expert-router coupling loss, a lightweight auxiliary loss that couples expert capabilities (activation norm) and the router’s decisions. The authors claim that this loss encourages each expert and each proxy token to match with each other, improving performance.

**Strengths:**

1. The ERC loss is computationally cheap.
2. The experiments show MoE gets considerable gain from this ERC loss.
3. Much analysis and ablation are provided.

**Weaknesses:**

1. You might need to compare with Router Orthogonalization Loss in https://yiyan.baidu.com/blog/publication/ERNIE_Technical_Report.pdf, since your loss is somewhat similar to ||(RW_g)TRWg - I||_F, if you assume W_g^TW_g\approx I, it is similar to ||R^TR - I||_F.
2. It seems that this ERC loss can be optimized to 0 when RMS(R) -> 0 or RMS(W_g) -> 0, so will it only serve like weight decay?

**Questions:**

1. Are both R and W_g optimized by ERC loss (rather than R only)?
2. Can you directly calculate the expectation of ERC loss under \delta and optimize the expectation directly?

---

> ### Author Response · Authors · 2025-11-21
> **Response to Reviewer WsdJ (1)**
>
> We sincerely appreciate your valuable comments, and hope our responses and revisions address your concerns.
>
> ---
> # W1: Compare with router-orth loss
> First, our loss can never be reduced to the router-orth loss. One could, in principle, have orthogonal experts and an orthogonal router ($R^T R = I$) while still suffering from a complete decoupling: for example, if each $R[i]$ is aligned with $W_g^{j \neq i}$. In this case, the router-orth loss is 0, but the model would be non-functional because the router always activates the wrong expert. Our ERC loss specifically penalizes this mismatch between the router's decisions and the experts' actual abilities, while the router-orth loss fails to address this issue.
>
> Second, we show that router-orth loss provides limited performance gains:
> ||ARC-C|CommonQA|COPA|BoolQ|Hella|OpenbookQA|SciQ|SociQA|MMLU|Wino|AVG|
> |-|-|-|-|-|-|-|-|-|-|-|-|
> |MoE|38.12|43.98|77.00|60.64|63.71|39.00|89.20|45.24|34.82|62.19|**55.39**|
> |+ERC|36.79|45.94|80.00|61.32|64.18|38.40|90.60|45.65|36.56|61.25|**56.07**|
> |+Orth|33.11|43.16|81.00|63.67|64.01|37.60|90.10|46.67|35.81|60.30|**55.54**|
>
> We also found that the router embeddings in our baseline MoE model are already nearly orthogonal, with an average absolute cosine similarity of 0.15. This value corresponds to angles between router embeddings mostly ranging from arccos(0.15) = 81° to arccos(-0.15) = 99°. Notably, we do not imply that all MoEs always have nearly orthogonal router embeddings, as this may depend on the data or specific architecture; we report this only as a characteristic of models we trained, which explains the limited gains from the Orth loss.
>
> This result confirms that weak router-expert coupling is a more critical issue than imperfect orthogonality: *our ERC loss provides significant gains even when applied to a baseline with already near-orthogonal routers.*
>
> In summary:
> 1. ERC loss does not degenerate into geometric constraints applied to experts or routers separately (e.g., the Orth loss).
> 2. The downstream performance shows that the issue we address is more critical, yet it has been largely overlooked.
>
> We have added this experiment, along with the performance plot evaluated throughout the training, to the updated PDF (Appendix C.3).
>
> ---
> # W2: Does the loss only act as weight decay
> No. If RMS(R\)→0 (a near-zero matrix), the routing logits would become uniform. This would cause the router to consistently select the first K experts by index, leading to severe load imbalance. However, as shown in Figure 3b, this does not happen. Similarly, RMS (Wg)→0 would make experts output near-zero activations, losing their capacities, which also does not happen.
>
> We provide direct evidence from layer-wise parameter norms (evaluated on the last checkpoint). As shown in Tab 1&2, the parameter norms are similar in magnitude. However, the vanilla MoE yields a very high ERC loss in Tab 3, demonstrating that ERC loss provides meaningful coupling.
>
> Tab 1: Expert Norm ($\Vert W^i_g\Vert$)
> |Layer|+ERC|Vanilla MoE|
> |-|-|-|
> |0|24.14$\pm$3.02|25.46$\pm$3.93|
> |1|29.42$\pm$0.69|30.14$\pm$0.68|
> |2|29.88$\pm$0.76|30.63$\pm$0.77|
> |3|29.42$\pm$0.78|30.18$\pm$0.77|
> |4|29.88$\pm$1.09|30.59$\pm$1.21|
> |5|29.86$\pm$1.06|30.33$\pm$1.13|
> |6|29.82$\pm$1.11|30.65$\pm$1.15|
> |7|29.96$\pm$1.16|30.56$\pm$1.20|
> |8|29.82$\pm$0.88|30.46$\pm$1.02|
> |9|29.86$\pm$0.79|30.58$\pm$0.88|
> |10|30.16$\pm$0.89|30.80$\pm$1.03|
> |11|31.50$\pm$1.26|32.03$\pm$1.46|
>
> Tab 2: Router Norm ($\Vert R[i]\Vert$)
> |Layer|+ERC|Vanilla MoE
> |-|-|-|
> |0|1.67±0.31|1.85±0.39|
> |1|1.13±0.12|1.25±0.13|
> |2|1.07±0.09|1.17±0.12|
> |3|1.01±0.07|1.10±0.08|
> |4|0.89±0.05|1.03±0.08|
> |5|0.87±0.06|0.93±0.08|
> |6|0.83±0.07|0.86±0.08|
> |7|0.75±0.06|0.82±0.07|
> |8|0.76±0.06|0.77±0.06|
> |9|0.74±0.06|0.80±0.07|
> |10|0.69±0.06|0.74±0.08|
> |11|0.73±0.10|0.80±0.14|
>
> Tab 3: ERC Loss
> |Layer|+ERC|Vanilla MoE|
> |-|-|-|
> |0|0|0.87|
> |1|0|0.42|
> |2|0|0.45|
> |3|0|0.25|
> |4|0|0.28|
> |5|0|0.24|
> |6|0|0.22|
> |7|0|0.21|
> |8|0|0.15|
> |9|0|0.16|
> |10|0|0.21|
> |11|0|0.50|
>
> We have added these results in the updated PDF (Appendix C.5).

---

> > ### Author Response · Authors · 2025-11-21
> > **Response to Reviewer WsdJ (2)**
> >
> > # Q1: Are both R and W_g optimized by ERC loss
> > Yes
> >
> > ---
> > # Q2: Can you directly optimize the expectation of ERC loss under δ
> >
> > The expectation of a ERC loss term can be written as:
> > $$E[max(0, M[j,i] - M[i,i])]
> > = E_{\delta_i, \delta_j}[max(0, \Vert R[j]\odot\delta_j\Vert\Vert W^j_{g}\Vert\cos\theta_{i,j} - \Vert R[i]\odot\delta_i\Vert\Vert W^i_{g}\Vert\cos\theta_{i,i})]$$
> > Deriving a closed-form solution for this expectation is challenging because: (1) the max operator is non-linear, and (2) the distribution of $δ_i$ is conditioned on R[i] and its nearest neighbor, not an arbitrary j. We would be happy to further explore any suggestions you may have on this matter!
> >
> > ---
> > We sincerely thank you for your valuable question regarding the parameter norms. It prompted us to include clarifications in the paper to prevent potential misunderstandings. We are also grateful for your suggestion to add more baselines, thereby making our study more robust. Should any further concerns remain, we would be happy to discuss them.

---

### Official Review · Reviewer_2vnc · 2025-11-02

**Soundness:** 3
**Presentation:** 3
**Contribution:** 3
**Rating:** 8
**Confidence:** 4

**Summary:**

In this paper, to mitigate performance degradation caused by the representation mismatch between the router and experts in MoE architectures, each row of the router matrix is treated as a representative vector for the vectors processed by an expert. A constraint is introduced in the form of a loss function, ensuring that when values in the vicinity of this representative are input to the corresponding expert, the activation is maximized. This constraint helps ensure that the expert selected by the router based on the actual input is the one most efficiently activated by that input. Experimental results show that the proposed method consistently outperforms the standard MoE setup, with few exceptions.

**Strengths:**

Many recent MoE papers focus on improving routing. Methods exist to resolve the mismatch between routing and experts, such as adding an auxiliary loss to teach desired properties in expert specialization, or modifying the model architecture, like AoE (which is also adopted as a baseline in this paper). The proposed method belongs to the former category; it achieves its goals regarding expert specialization by simply adding a simple constraint, without modifying the conventional model architecture at all. The proposed method can leverage existing software assets as-is, for example, by being integrated into existing training toolkits. The proposed method is an extremely lightweight loss function, and it is believed to have no or very small practical impact on training speed when introduced into the MoE training process. Personally, I am impressed that the authors conceived of this method, and I would like to try it in our own training framework.

It is noteworthy that the proposed method enables the router and experts to have an explicit geometric correspondence, succeeding in achieving a similar effect to AoE without requiring significant architectural modifications. It may also facilitate the visual analysis of the model's internals.

**Weaknesses:**

The experiments only validate the method on a single, very small-scale model instance. It has not been demonstrated whether the method is effective across the wide variety of MoE architectures. Since the experiments involve expensive pre-training, it is understandable that validating on various settings must be forgone due to cost, but it is true that the information provided feels somewhat insufficient.

The method includes randomness, which may be a source of training instability, although as shown in 4.4 (2), the fact that this randomness contributes to generalization appears to be valid.

A new hyperparameter, $\alpha$, which is difficult to tune intuitively, is introduced. Given the current lack of experimentation across a wide range of model instances, applying the settings used in the paper directly to other experiments is considered to carry a certain amount of risk.

**Questions:**

Regarding the randomness: could a method be devised to make the training behavior more theoretically consistent and predictable? For example, could a derivative algorithm be considered, such as marginalizing $R[i] \odot \delta_i$ over $\delta_i$, or handling the noise as a distribution (without sampling)?

The proposed method can be seen as a form of contrastive learning between the router and expert features, and therefore it seems relatively natural to consider leveraging techniques from contrastive learning. Are there any thoughts on this at this time?

---

> ### Author Response · Authors · 2025-11-21
> **Response to Reviewer 2vnc (1)**
>
> We sincerely appreciate your valuable comments. We hope our responses and revisions address your concerns.
>
> ---
> # W1&W2: 3B-models are small & Potential instability caused by randomness
>
> We have started training larger models since the initial submission. Specifically, we scaled the architecture to 256 experts (keeping top-k=8) and doubled the model depth, resulting in a 15B parameter model with approximately 700M activated parameters. This model was trained for 500B tokens with $\alpha=1$. Here are results on some widely-used challenging public benchmarks (3-shot eval for BBH and MATH, and 5-shot for all others):
>
> |15B LLM|MMLU|CEVAL|MMLU-PRO|AGI-EVAL|BBH|MATH|GSM8k|TriviaQA|
> |-|-|-|-|-|-|-|-|-|
> |Baseline|63.2|67.5|31.0|42.0|44.3|25.7|45.2|47.2|
> |+ERC Loss|64.6|69.0|31.9|44.2|45.6|26.1|45.8|49.1|
>
> Moreover, throughout this large-scale training, we observed no loss spikes, indicating that our method maintains stability at this scale.
>
> We have added this experiment in the updated PDF (Section 4.3 & Table 1).
>
> ---
> # W3: $\alpha$ is difficult to tune intuitively.
>
> Here we present an intuitive principle for tuning α based on the sparsity of the MoE layer. The core idea is that an appropriate level of expert specialization (controlled by α) depends on sparisity (K/n).
>
> **Intuition:** When the MoE is very sparse (with a small K/n), the selected combination of experts must be generalist enough to cover the diverse requirements of processing any given token. Over-specialization (a small α) risks that this small set of experts cannot adequately handle the input, thereby hurting performance. Conversely, when K/n is large, the system can afford to include more specialized experts, as their collective capacity is more likely to cover the input's needs.
>
> **Validation:** We pre-trained models with n=64 experts, varying $K\in${4,8,16} and $\alpha\in${0.4,0.6,0.8,1.0}. Due to limited time, we trained 100B tokens for each $(K,\alpha)$ pair. All other settings follow the paper, and we report the average downstream score. The results confirm the intuition: for $K$=4 and 8, $\alpha$=1.0 performs best; while $\alpha$=0.6 is acceptable for $K$=16.
> |α/K|4|8|16|
> |-|-|-|-|
> |0.4|51.12|51.96|52.92|
> |0.6|51.49|52.07|53.32|
> |0.8|51.66|52.18|53.28|
> |1.0|51.68|52.31|53.39|
>
> **Practical Guideline:** Given that industrial MoEs operate with high sparsity (e.g., $K/n \ll 8/64$), we recommend using $\alpha=1$ as a robust default, requiring no further tuning. For research on smaller models or denser activations, $\alpha=1.0$ remains a safe and convenient choice, while $\alpha<1$ may yield more benefits but requires case-specific tuning. This experiment also shows that the ERC loss could serve as a tool for studying specialization.
>
> We have added this experiment in the updated PDF (Section 4.5 & Figure 7).

---

> > ### Author Response · Authors · 2025-11-21
> > **Response to Reviewer 2vnc (2)**
> >
> > # Q1: What about using the marginal distribution of $R[i]\odot\delta_i$?
> >
> > We appreciate this insightful question. Our method requires the $\tilde{R}$ to be centered on $R[i]$. For both uniform and Gaussian noise $\delta_i$ with zero mean, $E_{\delta_i}[R[i]\odot\delta_i]$ is $R[i]$ itself. As our ablation studies demonstrate, directly using $R[i]$ to compute the ERC loss leads to suboptimal performance.
> >
> > We interpret this question as probing whether we could minimize the ERC loss by optimizing its expectation directly:
> > $$E[max(0, M[j,i] - M[i,i])]
> > = E_{\delta_i, \delta_j}[max(0, \Vert R[j]\odot\delta_j\Vert\Vert W^j_{g}\Vert\cos\theta_{i,j} - \Vert R[i]\odot\delta_i\Vert\Vert W^i_{g}\Vert\cos\theta_{i,i})]$$
> > Deriving a closed-form solution for this expectation is challenging because: (1) the $\max$ operator is non-linear, and (2) the distribution of $δ_i$ is conditioned on $R[i]$ and its nearest neighbor, not an arbitrary $j$. We would be happy to further explore any suggestions you may have on this matter!
> >
> > ---
> > # Q2: Any thoughts on leveraging techniques from contrastive learning?
> >
> > Thank you for this interesting question. Although our method is fundamentally different from  a CL method applied solely to routers or experts, we have conducted an experiment inspired by the InfoNCE loss:$$L=-log\frac{exp(z^+/\tau)}{\sum_z exp(z/\tau)}$$We tried to reformulate the ERC loss into a similar structure:$$L_{ERC}=-\frac{1}{n^2}\sum_{i}^{n}(Y[i,:]\text{LogSoftmax}(M[:,i])+\text{LogSoftmax}(M[i,:])Y[:,i]),$$where the matrix $Y$ is:
> > $$Y[i,i]=1-\tau;Y[i,j\neq i]=\frac{\tau}{n-1}.$$However, this formulation is ineffective:
> > |τ|*Peak* Avg Score|
> > |-|-|
> > |0|56.2|
> > |0.3|55.3|
> > |0.5|55.8|
> >
> > For reference, the vanilla MoE baseline peaks at 55.9, and our original method reaches 56.7. The performance plot for this new formulation was also consistently low across training. This formulation fails for two main reasons:
> >
> > 1.  *Low Flexibility*: Our original method only requires $M[i,j] < M[i,i]$. It places no constraints on the relationship between $M[i,j]$ and $M[i,k]$ for any $j,k\neq i$, allowing the model the freedom to explore the relationship between these (i.e., $j,k\neq i$) experts and routers. The new formulation, in contrast, forces all off-diagonal entries M[i, j] to be similar, which overly constrains the model and reduces this flexibility.
> >
> > 2.  *Difficulty in Hyperparameter Tuning*: The temperature parameter τ is considerably more difficult to tune than our $\alpha$. First, no value of τ could make the loss equivalent to our method when $\alpha=1$. Second, finding the optimal τ is complicated by its sensitivity to the magnitude of router and expert weights, which can vary across layers.
> >
> > In summary, while leveraging CL techniques is a valuable direction, we have so far not yielded further improvements. We believe our analysis and discussion here could provide helpful insights for future work, and we thank you again for this interesting question.
> >
> > ---
> > We sincerely thank you for your constructive suggestions and valuable comments, which improved this paper. We also appreciate your positive feedback on our work. Should any further concerns remain, we would be happy to discuss them.

---

### Official Review · Reviewer_n6Ei · 2025-11-03

**Soundness:** 4
**Presentation:** 4
**Contribution:** 3
**Rating:** 6
**Confidence:** 2

**Summary:**

This paper proposes a new auxiliary loss for MoE training which better couples the experts and the routers and promotes expert specialization. The idea is to think of the router rows as cluster centers and to generate cluster data from the centers via random perturbations. The data from a given cluster should induce higher activation norm in the corresponding expert relative to the other experts and to data from other clusters. This is enforced by a soft hinge penalty. Adding the auxiliary loss adds modest overhead during training and does not affect inference. The authors find that the penalty improves downstream metrics over a baseline vanilla MoE and is comparable to the more expensive AoE method.

**Strengths:**

The method is simple, intuitive, and clearly presented. The experiments and evaluations are thorough and the auxiliary loss does appear to improve performance.

**Weaknesses:**

The activation metric is not scale invariant, the auxiliary loss can be decreased in a non-meaningful manner simply by scaling up $W_g^i$.

The auxiliary appears to make the gradient dense across experts since activations norms for each token are computed across experts.

**Questions:**

In addition to clarification of the weaknesses, I have the following questions:

Is it possible that $\alpha > 1$ can perform even better? At what $\alpha$ will you recover vanilla MoE?

What about using post SwiGLU activations or $W_o$?

Do the norms $\lVert R[i] \rVert$ stay comparable across $i$? If this is not true, then there seems to be a mismatch between Euclidean distance based clustering and inner-product based routing.

---

> ### Author Response · Authors · 2025-11-21
> **Response to Reviewer n6Ei (1)**
>
> We sincerely appreciate your valuable comments. We hope our responses and revisions address your concerns.
>
> ---
> # W1: Can the loss be decreased in a non-meaningful way simply by scaling up $\Vert W^i_g\Vert$?
>
> (We assume you mean scaling up the norm $\Vert W^i_g\Vert$, rather than scaling the matrix $W^i_g$ itself. If our understanding is incorrect and you indeed mean scaling $W^i_g$, please let us know so we can provide a more targeted discussion. Thank you!)
>
> No. The structure of the ERC loss inherently prevents this, as any attempt to reduce one loss term by increasing a norm will simultaneously increase other terms in the loss function.
>
> Here is a detailed explanation:
>
> The term M[i,j] can be expressed as $\Vert R[i]\Vert\Vert W^j_{g}\Vert\cos\theta_{i,j}$, where $\theta_{i,j}$ denotes the angle between $R[i]$ and $W^j_{g}$.
>
> Scaling up $\Vert W^i_{g}\Vert$ decreases the loss from $i$-th row in M (as the second term below increases):
> $\Vert R[i]\Vert\Vert W^j_{g}\Vert\cos\theta_{i,j} - \Vert R[i]\Vert\Vert W^i_{g}\Vert\cos\theta_{i,i}$.
>
> However, simultaneously, it increases the loss term from every $j \neq i$ rows (because the first term below increases):
> $\Vert R[j]\Vert\Vert W^i_{g}\Vert\cos\theta_{j,i} - \Vert R[j]\Vert\Vert W^j_{g}\Vert\cos\theta_{j,j}$.
>
> To further address concerns regarding parameter norms, we compare the Frobenius norms (mean ± standard deviation) of the expert matrices $W^{i}_g$ (across all experts i) between our model and the vanilla MoE baseline for each layer. The results show that, with the ERC loss, the norms actually tend to decrease slightly, contrary to the concern that they are trivially increased.
>
> |Layer|w. ERC loss|w/o. ERC loss|
> |-|-|-|
> |0|24.14$\pm$3.02|25.46$\pm$3.93|
> |1|29.42$\pm$0.69|30.14$\pm$0.68|
> |2|29.88$\pm$0.76|30.63$\pm$0.77|
> |3|29.42$\pm$0.78|30.18$\pm$0.77|
> |4|29.88$\pm$1.09|30.59$\pm$1.21|
> |5|29.86$\pm$1.06|30.33$\pm$1.13|
> |6|29.82$\pm$1.11|30.65$\pm$1.15|
> |7|29.96$\pm$1.16|30.56$\pm$1.20|
> |8|29.82$\pm$0.88|30.46$\pm$1.02|
> |9|29.86$\pm$0.79|30.58$\pm$0.88|
> |10|30.16$\pm$0.89|30.80$\pm$1.03|
> |11|31.50$\pm$1.26|32.03$\pm$1.46|
>
> In summary, the model cannot "hack" the erc loss through increasing or decreasing parameter norms. The overall reduction of ERC loss is only achieved by meaningfully coupling router embeddings and respective experts.
>
> We have added related discussions in the updated PDF (Appendix C.5).
>
> ---
> # W2: The loss appears to make the gradient dense across experts since activations norms for each token are computed across experts.
>
> Thank you for this comment.
>
> If your concern was that we feed *every real input token to every expert*, we hope to clarify some misunderstanding. The ERC loss is computed using a set of **proxy tokens**, not the model's actual input tokens. Specifically, we construct **n** proxy tokens by adding noise to the **n** router embeddings. Each of these proxy tokens is then processed by **n** experts to calculate their activation norms.
>
> If your concern is indeed about the fact that *each proxy token is processed by every expert*, we would be grateful for your further elaboration on why this might be problematic from a gradient perspective. We look forward to your insightful thoughts and are happy to discuss it further.

---

> > ### Author Response · Authors · 2025-11-21
> > **Response to Reviewer n6Ei (2)**
> >
> > # Q1: Performance with $\alpha>1$
> > > At what $\alpha$ will you recover vanilla MoE?
> >
> > This question seeks the minimum α that zeroes the ERC loss for a pre-trained vanilla MoE's M matrix. The table shows the "post-hoc" ERC loss evaluation of the vanilla MoE across $\alpha$ values (evaluated on the last checkpoint). Achieving zero ERC-loss across all layers demands $\alpha=5$. This provides direct evidence that the router-expert coupling in the vanilla MoE is very weak.
> >
> > |Layer|$\alpha=1$|$\alpha=2$|$\alpha=3$|$\alpha=4$|$\alpha=5$|
> > |-|-|-|-|-|-|
> > |0|0.87|0.69|0.26|0|0|
> > |1|0.42|0.28|0.10|0|0|
> > |2|0.45|0.19|0|0|0|
> > |3|0.25|0.15|0|0|0|
> > |4|0.28|0.08|0|0|0|
> > |5|0.24|0.22|0|0|0|
> > |6|0.22|0.15|0|0|0|
> > |7|0.21|0.13|0|0|0|
> > |8|0.15|0.05|0|0|0|
> > |9|0.16|0|0|0|0|
> > |10|0.21|0.09|0|0|0|
> > |11|0.50|0.44|0.20|0.20|0|
> >
> > Note that for a clear and concise demonstration, the loss values in this table are computed using the original $R$ rather than $\tilde{R}$, making the results deterministic.
> >
> > > Can $\alpha$>1 perform better?
> >
> > When $\alpha>1$, it contradicts our core motivation, as the router and experts shift from a state of no mismatch to looser coupling constraints, ultimately causing the model to degenerate into a vanilla MoE.
> >
> > To verify this, we pre-trained two additional models with the ERC loss at $\alpha$ values of 2 and 3. We found that at $\alpha=2$, the model showed only limited improvement, and at $\alpha=3$, the model showed almost no improvement over the vanilla MoE. Because the last checkpoint's score cannot reflect this trend, we provide the full performance plots in Appendix C.4 rather than listing a table here.
> >
> > ---
> > # Q2: What about using post SwiGLU activations or $W_o$?
> > We appreciate this suggestion. While we initially excluded these variants due to their computational overhead, we have now trained two new models using them respectively. Our results show that using the post-SwiGLU variant is ineffective, while using the activation after $W_o$ performs second-best among all five candidates. Considering the overhead, our original method is more cost-effective.
> >
> > The comparison table at the final checkpoint is presented below. While the table might suggest that the post-SwiGLU variant is not particularly underperforming, the performance plot tracked throughout the entire training process (included in Appendix C.1 of the updated PDF) demonstrates that it in fact provided insignificant gains.
> >
> > ||ARC-C|CommonQA|COPA|BoolQ|Hella|OpenbookQA|SciQ|SociQA|MMLU|Wino|AVG|
> > |-|-|-|-|-|-|-|-|-|-|-|-|
> > |Baseline|38.12|43.98|77.00|60.64|63.71|39.00|89.20|45.24|34.82|62.19|**55.39**|
> > |$W_g$(Ours)|36.79|45.94|80.00|61.32|64.18|38.40|90.60|45.65|36.56|61.25|**56.07**|
> > |Post-SwiGLU|34.78|41.68|84.00|62.90|64.00|38.20|89.70|45.08|36.65|60.22|**55.72**|
> > |$W_o$|35.79|44.30|86.00|62.02|63.75|37.40|90.20|45.08|35.33|59.66|**55.95**|
> >
> > ---
> > # Q3: Do the norms $\Vert R[i]\Vert$ stay comparable across i?
> >
> > Yes, the overall ERC loss is minimized only when the model keeps the router embedding norms comparable. Here are the reasons:
> >
> > As also discussed in our response to Weakness 1, if an arbitrary $\Vert R[i]\Vert$ increases, although the loss for the i-th column decreases (as the second term below becomes larger):
> > $$\Vert R[j]\Vert\Vert W^i_g\Vert\cos\theta_{j,i}-\Vert R[i]\Vert\Vert W^i_g\Vert\cos\theta_{i,i}.$$
> > It simultaneously increases the loss contribution from every other column $j \neq i$ (as the first term below becomes larger):
> > $$\Vert R[i]\Vert\Vert W^j_g\Vert\cos\theta_{i,j}-\Vert R[j]\Vert\Vert W^j_g\Vert\cos\theta_{j,j}$$
> > A similar effect happens when an arbitrary $\Vert R[i]\Vert$ decreases.
> > This property of the ERC loss inherently regularizes the router embedding norms to be comparable.
> >
> > As shown in the table below, our model exhibits a smaller standard deviation than the vanilla MoE across layers, confirming that $\Vert R[i]\Vert$ values are more consistent when trained with the ERC loss.
> >
> > |Layer|w. ERC|w/o. ERC|
> > |-|-|-|
> > |0|1.67$\pm$0.31|1.85$\pm$0.39|
> > |1|1.13$\pm$0.12|1.25$\pm$0.13|
> > |2|1.07$\pm$0.09|1.17$\pm$0.12|
> > |3|1.01$\pm$0.07|1.10$\pm$0.08|
> > |4|0.89$\pm$0.05|1.03$\pm$0.08|
> > |5|0.87$\pm$0.06|0.93$\pm$0.08|
> > |6|0.83$\pm$0.07|0.86$\pm$0.08|
> > |7|0.75$\pm$0.06|0.82$\pm$0.07|
> > |8|0.76$\pm$0.06|0.77$\pm$0.06|
> > |9|0.74$\pm$0.06|0.80$\pm$0.07|
> > |10|0.69$\pm$0.06|0.74$\pm$0.08|
> > |11|0.73$\pm$0.10|0.80$\pm$0.14|
> >
> > We hope this analysis addresses your concern, and we thank you for this insightful question. We have added these results in the updated PDF (Appendix C.5).
> >
> > ---
> > We sincerely thank you for your insightful and valuable questions. Your questions regarding the parameter norms inspired us to add a clarification in the paper to prevent potential misunderstandings. We also appreciate your suggestion to experiment with more activation variants.
> >
> > Regarding Weakness 2, we are not entirely sure if our response has fully captured the essence of your concern. Should our response have missed the mark, we would sincerely appreciate any additional comments.

---

> ### Comment · Reviewer_n6Ei · 2025-11-27
>
> Thank you for addressing the review. The only remaining point I would like to clarify is
>
> W2: I don't view it as a problem in terms of gradients, but I'm curious about the computational overhead. Conventional MoE training doesn't incur the same overhead since the gradients are sparse.

---

> > ### Author Response · Authors · 2025-11-28
> >
> > Thank you for the clarification. We hope this response could address your concerns.
> >
> > We provide a starightforward explanation of the computational overhead associated with our ERC loss, to complement the FLOPs analysis in the paper.
> >
> > Notably, the computation of ERC loss involves:
> > 1.  Generating a noisy version of each router embedding.
> > 2.  Passing all $n$ noised router embeddings through all $n$ experts to compute post-$W_g$ activations, yielding $n^2$ activations in total.
> >
> > The computational cost for this process is approximately equivalent to a forward pass for $n^2 / 3$ tokens. *For these noised embeddings*, all experts are activated, *but for each expert*, activation remains sparse in the broader context of the forward pass.
> >
> > For a practical analysis using data parallelism and expert parallelism, we use our newly-added 15B model with the configuration (Top-$K$=8, $T$=3M, $n$=256, dp_size=64, ep_size=8). The cost per device is:
> > *   Base MoE forward: $K * T$ / dp_size
> > *   ERC Overhead: $n$ * ($n$ / ep_size) / 3
> >
> > Therefore, the ERC overhead constitutes only 0.72% of the base model's forward cost. In practice, our measured throughput of 62.03B tokens per day for the baseline versus 61.52B tokens per day for our model (a 0.82% reduction) is consistent with the theoretical estimate, confirming the overhead is marginal. With a smaller $n=64$, as in our 3B models trained with dp_size=32 and ep_size=1 (i.e., EP disabled), the overhead ratio drops further to 0.18\%.
> >
> > We have included the empirical training overhead and measured throughputs in Appendix B.2 of the updated PDF.

---

### Comment · Area_Chair_GC3E · 2025-11-27
**Rebuttal and Discussion Phase**

Dear Reviewers,

Thank you again for your time and effort in reviewing this paper. We are approaching the discussion deadline. I kindly ask you to review the rebuttal and continue the discussion so that we can reach a well-considered decision.

---

### Meta-Review · Area_Chair_F26W · 2026-01-06

**Summary:**

All three reviews agree that the paper proposes a simple, intuitive, and practically relevant auxiliary loss (ERC) that explicitly couples MoE experts with the router, and that the empirical results show consistent gains over a strong vanilla MoE baseline with very low overhead. The main concerns are not about correctness, but about:

**Objective properties and optimization behavior (Reviewer n6Ei, WsdJ):**
- The activation-based metric is not scale invariant; in principle the loss might be reduced in non-meaningful ways (e.g., scaling norms of router or expert weights).
- ERC may densify gradients across experts because it computes activation norms across all experts.
- Clarification was needed on whether ERC is effectively acting like weight decay or router orthogonalization.

**Scope and robustness of empirical validation (Reviewer 2vnc):**
- Initial experiments focused on a single 3B MoE-LLM setup, leaving open how well ERC generalizes to larger models and different MoE configurations.
- ERC introduces randomness (via noisy proxy tokens) and a new hyperparameter α\alphaα; reviewers wanted more evidence on stability and guidelines for tuning.

**Comparisons and relation to prior work (Reviewer WsdJ):**
- The relationship to router-orthogonalization losses (e.g., ERNIE technical report) needed to be clarified, both conceptually and empirically.

Despite these concerns, all three reviewers already scored the paper above the acceptance threshold (two 6’s and one 8), viewing it as a useful and well-motivated contribution to MoE training.

**Reviewer Concerns:**

The rebuttal and revision address the key technical and empirical concerns quite convincingly:

- **Scale-invariance / “hackability” of the loss (n6Ei, WsdJ):** The authors provide a detailed analysis showing that scaling one expert’s or router’s norm reduces some terms in the ERC loss but simultaneously increases others, so the loss cannot be trivially minimized by norm inflation or collapse. They also report layer-wise Frobenius norms for expert and router weights, showing that norms remain comparable between baseline and ERC models, while ERC loss values differ substantially. This strongly supports the claim that ERC is not merely acting as weight decay and that the model cannot game the loss by rescaling parameters.

- **Gradient density and computational overhead (n6Ei):** The authors clarify that ERC is computed on proxy tokens, not actual training tokens, and quantify the overhead both theoretically and empirically. For a 15B model with realistic data/expert parallelism, ERC adds ~0.7–0.8% training cost, with measured throughput decreasing from 62.03B to 61.52B tokens/day. For smaller models, overhead is even lower. This directly addresses concerns about practicality.

- **Experimental scope and stability (2vnc):** The authors add a 15B-parameter MoE-LLM experiment (256 experts, deeper model, 500B tokens), showing consistent improvements across a wide set of challenging benchmarks (MMLU, CEVAL, MMLU-PRO, AGI-Eval, BBH, MATH, GSM8k, TriviaQA), with no loss spikes, indicating stability at scale. They further provide a principled $\alpha$-tuning guideline based on MoE sparsity $K/n$, supported by ablations over multiple $K$ and $\alpha$ combinations.

- **Relation to router-orthogonalization and additional baselines (WsdJ):** The authors both conceptually and empirically distinguish ERC from router-orth losses. They show that even when router embeddings are already nearly orthogonal, ERC still provides meaningful gains, and a router-orth loss yields only minor improvements. This supports their claim that weak router–expert coupling, not router non-orthogonality, is the key issue ERC addresses.

- **Randomness & expectation over noise, contrastive variants (2vnc, WsdJ):** The rebuttal discusses why directly optimizing the expectation over noise is non-trivial and reports experimental attempts to recast ERC as an InfoNCE-style contrastive loss, which underperform the original formulation. This shows the authors have seriously explored these directions, even if a closed-form expectation or a better contrastive variant remains open.

**Outstanding (but minor) concerns:**

What remains is mostly scope rather than correctness: ERC is demonstrated on one primary architecture family (MoE-LLMs) and a set of sizes (3B, 15B), and while these are strong and realistic experiments, there is naturally more space to explore other MoE flavors and tasks. Also, tuning $\alpha$ is now better understood but still not completely “automatic.” These are reasonable limitations for a first paper on ERC and do not undermine the main contribution.

Overall, the rebuttal effectively resolves the central technical and empirical doubts and reinforces the case for acceptance.

**Reviewer Scores:**

**Reviewer n6Ei (initial rating: 6):** Reviewer n6Ei's main worries were about non-meaningful loss reduction via scaling and gradient density / overhead. The authors provide strong analytical and empirical evidence that norms do not blow up or collapse and that overhead is negligible. After the rebuttal, Reviewer n6Ei acknowledges the clarifications and raises only one follow-up about computational cost, which is also answered in detail. I expect Reviewer n6Ei would maintain or slightly increase their score (e.g., remain at 6 or move to 7), staying clearly in the acceptance range.

**Reviewer 2vnc (rating: 8):**  Reviewer 2vnc was already strongly positive, mainly asking for evidence of scalability and guidance on randomness and $\alpha$. The added 15B experiments, stability observations, and tuning heuristic directly target these concerns. I expect Reviewer 2vnc would remain at 8 and continue to advocate for acceptance, possibly with higher confidence.

**Reviewer WsdJ (rating: 6):** Reviewer WsdJ raised important questions about similarity to router-orth losses and potential degeneration into weight decay. The authors’ new experiments and norm/ loss analyses address these concerns thoroughly, and they also added the requested router-orth baseline. I expect Reviewer WsdJ would remain at 6, or plausibly move to a 7, viewing ERC as a simple, effective auxiliary loss with clear empirical benefits and good analysis.

---

### Decision · Program_Chairs · 2026-01-26

Accept (Oral)